# Quality of Life as a Mediator between Cancer Stage and Long-Term Mortality in Nasopharyngeal Cancer Patients Treated with Intensity-Modulated Radiotherapy

**DOI:** 10.3390/cancers13205063

**Published:** 2021-10-10

**Authors:** Kuan-Cho Liao, Hui-Ching Chuang, Chih-Yen Chien, Yu-Tsai Lin, Ming-Hsien Tsai, Yan-Ye Su, Chao-Hui Yang, Chi-Chih Lai, Tai-Lin Huang, Shau-Hsuan Li, Tsair-Fwu Lee, Wei-Ting Lin, Chien-Hung Lee, Fu-Min Fang

**Affiliations:** 1Department of Radiation Oncology, Kaohsiung Chang-Gung Memorial Hospital and Chang Gung University College of Medicine, Kaohsiung 833401, Taiwan; piko@cgmh.org.tw; 2Department of Public Health, College of Health Sciences, Kaohsiung Medical University, Kaohsiung 807378, Taiwan; 3Department of Otolaryngologist, Kaohsiung Chang-Gung Memorial Hospital and Chang Gung University College of Medicine, Kaohsiung 833401, Taiwan; entjulia@cgmh.org.tw (H.-C.C.); cychien3965@adm.cgmh.org.tw (C.-Y.C.); xeye@cgmh.org.tw (Y.-T.L.); b9302094@cgmh.org.tw (M.-H.T.); yanyesu@cgmh.org.tw (Y.-Y.S.); chingmn@cgmh.org.tw (C.-H.Y.); gordon93@cgmh.org.tw (C.-C.L.); 4Department of Hematology and Oncology, Kaohsiung Chang-Gung Memorial Hospital and Chang Gung University College of Medicine, Kaohsiung 833401, Taiwan; victor99@cgmh.org.tw (T.-L.H.); lee0624@cgmh.org.tw (S.-H.L.); 5Medical Physics and Informatics Laboratory of Electronics Engineering, National Kaohsiung University of Science and Technology, Kaohsiung 80778, Taiwan; tflee@nkust.edu.tw; 6Department of Medical Imaging and Radiological Sciences, Kaohsiung Medical University, Kaohsiung 807378, Taiwan; 7Department of Social, Behavioral and Population Sciences, School of Public Health and Tropical Medicine, Tulane University, New Orleans, LA 70112, USA; lweiting@tulane.edu; 8Research Center for Environmental Medicine, Kaohsiung Medical University, Kaohsiung 807378, Taiwan; 9Department of Medical Research, Kaohsiung Medical University Hospital, Kaohsiung Medical University, Kaohsiung 807378, Taiwan; 10Office of Institutional Research & Planning, Secretariat, Kaohsiung Medical University, Kaohsiung 807378, Taiwan; 11Department of Medicine, Chang Gung University College of Medicine, Taoyuan 333323, Taiwan

**Keywords:** quality of life, mediator, nasopharyngeal carcinoma, intensity modulated radiotherapy, mortality, Baron and Kenny’s method

## Abstract

**Simple Summary:**

Even after the implementation of intensity-modulated radiotherapy (IMRT), nasopharyngeal cancer (NPC) survivors may continue to exhibit several physical symptoms that negatively affect long-term quality of life (QoL). An NPC patient cohort study (*n* = 682) was conducted to examine the potential mediating effect of QoL (evaluated at multiple treatment-related time points) on the cancer stage–mortality association. Patients with advanced NPC exhibited low global health QoL and high QoL-HN35 symptom pre-IMRT, 3 months post-IMRT, and 2 years post-IMRT. Global health QoL and QoL-HN35 symptom scores 2 years after IMRT explained 49.4% and 39.4% of the excessive effect of advanced NPC on mortality risk. Our findings indicate that global health QoL and QoL-HN35 symptom 2 years after IMRT are key mediators of the relationship between advanced NPC and high mortality. These findings emphasize the significance of QoL-HN35 symptom and global health QoL-associated medical support and care for patients with NPC who received IMRT.

**Abstract:**

Background: Quality of life (QoL) attained before, during, or after treatments is recognized as a vital factor associated with therapeutic benefits in cancer patients. This nasopharyngeal cancer (NPC) patient longitudinal study assessed the relationship among QoL, cancer stage, and long-term mortality in patients with nasopharyngeal carcinoma (NPC) treated with intensity-modulated radiotherapy (IMRT). Patients and Methods: The European Organization for Research and Treatment of Cancer (EORTC) core QoL questionnaire (QLQ-C30) and the head and neck cancer-specific QoL questionnaire module (QLQ-HN35) were employed to evaluate four-dimensional QoL outcomes at five time points: pre- (*n* = 682), during (around 40 Gy) (*n* = 675), 3 months (*n* = 640), 1 year (*n* = 578) and 2 years post-IMRT (*n* = 505), respectively, for 682 newly diagnosed NPC patients treated between 2003 and 2017 at a single institute. The median followed-up time was 7.5 years, ranging from 0.3 to 16.1 years. Generalized estimating equations, multivariable proportional hazards models, and Baron and Kenny’s method were used to assess the investigated effects. Results: Advanced AJCC stage (III–IV) patients revealed a 2.26-fold (95% CI—1.56 to 3.27) higher covariate-adjusted mortality risk than early-stage (I–II) patients. Compared with during IMRT, advanced-stage patients had a significantly low global health QoL and a significantly high QoL-HN35 symptom by a large magnitude at pre-, 3 months, and 2 years post-IMRT. QoL scales at pre-IMRT, 1 year, and 2 years post-IMRT were significantly associated with mortality. The effect changes of mortality risk explained by global health QoL, QoL-C30, and QoL-HN35 symptom were 5.8–9.8% at pre-IMRT but at 2 years post-IMRT were 39.4–49.4% by global health QoL and QoL-HN35 symptoms. Conclusions: We concluded advanced cancer stage correlates with a long-term high mortality in NPC patients treated with IMRT and the association is partially intermediated by QoL at pre-IMRT and 2 years post-IMRT. Therefore, QoL-HN35 symptom and global health QoL-dependent medical support and care should be focused and tailored at 2 years post-IMRT.

## 1. Introduction

In recent years, the medical application of intensity-modulated radiotherapy (IMRT) and optimization of chemotherapy strategies have prominently prolonged survival and lessened toxicities to normal tissues for patients with nasopharyngeal cancer (NPC) [1]. However, the 5-year survival rate for advanced-stage NPC (stages III and IV) is still at an ameliorable level, both in Asian areas (e.g., 23–79% in Japan and 60–79% in Taiwan) and European areas (e.g., 31–55% in Finland) [2,3,4]. Because the current anatomy-based staging system is insufficient for predicting therapeutic benefits or clinical prognosis [1], investigations that consider other clinical and non-clinical factors to help survival prediction are warranted.

Quality of life (QoL) attained before, during, or after treatments is recognized as a vital factor associated with therapeutic benefits in cancer patients [5,6,7,8,9,10,11]. Clinical investigations have demonstrated that even with the implementation of intensity-modulated radiotherapy (IMRT), NPC survivors still suffered several physical symptoms that influence long-term QoL [12,13]. Patients with advanced-stage NPC were identified as having poorer QoL than early-stage patients [13,14,15]. Alternatively, QoL has been considered a potential survival predictor for cancer patients. In longitudinal studies of NPC patients with QoL measured pretreatment and 1 year after treatment, an increase in physical functioning and a decrease in fatigue and appetite loss were identified to predict a higher overall survival [9,10].

Studies have reported that approximately 69.0–88.2% of patients with NPC have already suffered from stage III or IV tumors at diagnosis [11,16,17,18]. Because advanced-stage NPC was associated with a lower QoL score and an inferior level of QoL was linked to a higher mortality, this raised the issues of what time periods of QoL in the treatment and follow-up course and what types of QoL can mediate the effect of advanced cancer stage on long-term mortality for NPC patients treated with IMRT. Hence, the purpose of this longitudinal study of a patient cohort was to investigate the relationship among cancer stage, QoL, and long-term mortality in NPC patients treated with IMRT, and to evaluate the potential mediating effect of QoL evaluated at multiple treatment-related time points on the association of cancer stage with NPC patient mortality.

## 2. Patients and Methods

### 2.1. Patient Data

Pathology-confirmed NPC patients, newly diagnosed at the Kaohsiung Chang-Gung Memorial Hospital with previously untreated, non-recurrence, and non-distant metastatic status between April 2003 and December 2017 were enrolled. NPC patients with previous or synchronous malignancies (*n* = 8), metastasis at diagnosis (*n* = 15), incapable of completing the prescribed treatment course of IMRT (*n* = 19), and unable to complete the QoL questionnaires at the time point of pre-IMRT (*n* = 28) were excluded. A total of 682 consecutively recruited NPC patients aged 20–80 years participated in this longitudinal study. Among the participants, 65 (9.5%) and 167 (24.5%) had AJCC stage I and II, and 244 (35.8%) and 206 (30.2%) had AJCC stage III and IV, determined by the 8th American Joint Committee on Cancer (AJCC) staging system [19].

### 2.2. Treatment

The technical details of IMRT in the institute during the study period have been presented in previous reports [9,20]. Before 2011, seven-fixed-beam IMRT with two-phase sequential planning was applied in 436 (63.9%) patients. The median prescribed dose to the gross tumor and nodal area was 70.2 Gy (range 62.4–78.6 Gy) and to the elective risky area was 45.0–59.4 Gy, with a daily fraction of 1.8 or 2.0 Gy and five fractions per week. After 2011, rotational-arc IMRT with simultaneously integrated boost planning was applied in 246 (36.1%) patients, with three dose levels of 69.96, 59.4, and 52.8 Gy in 33 fractions prescribed to the high, intermediate, and low clinical target area, respectively. Cisplatin-based chemotherapy was given weekly, intravenously, during the course of IMRT as a radiation sensitizer for those with clinical stages II–IV. Neoadjuvant or adjuvant chemotherapy with the combination regimens of cisplatin and 5-fluorouracil administered every 3–4 weeks was given for 1–4 cycles to those patients with clinical stages III–IV or receiving inadequate doses of cisplatin during the course of IMRT.

### 2.3. Follow-Up

In our treatment network, NPC patients were scheduled to visit the tumor clinics every 3 months after IMRT for primary assessment of treatment effect in the first 2 years, and for regular image and physical examinations every 4–6 months in the third to fifth years and bi-annually in the fifth to tenth years, respectively. After 10 years, patients were followed up year after year for their health condition in the clinics or through telephone interviews. Health insurance data, medical charts, and death certificates were used to determine the vital status and the date of death where relevant for each patient. The follow-up of the patient cohort started on 1 April 2003 and finished on 31 December 2019 (censoring date). Time at risk was defined as the time from the date of tumor stage diagnosis to the date of death or censoring date. The median followed-up time was 7.5 years, ranging from 0.3 to 16.1 years.

### 2.4. QoL Instruments 

The core QoL questionnaire (QLQ-C30) and the head and neck cancer-specific QoL questionnaire module (QLQ-HN35) developed by the European Organization for Research and Treatment of Cancer (EORTC, Brussels, Belgium), Taiwan Chinese version 3, were used to collect QoL data from each patient [21]. EORTC QLQ-C30 comprises a scale for global health status, five scales for multidimensional functioning (physical, cognitive, role, emotional, and social functioning), and nine scales/items for disease symptoms (fatigue, nausea, and vomiting, pain, dyspnea, insomnia, diarrhea, constipation, appetite loss, and financial difficulties) [22]. EORTC QLQ-HN35 includes seven symptom scales (problems of pain, swallowing, speech, senses, social contact, social eating, and sexuality), six symptom items (trouble with opening the mouth, teeth, sticky saliva, dry mouth, coughing, and feeling ill), and five dichotomous items (use of painkillers, nutritional supplements, and feeding tube, and weight loss or weight gain). Except for five QLQ-HN35 dichotomous items and one QLQ-C30 seven-point global health scale, all scales/items were structured in a four-point response. The responses were linearly converted to a score between 0 and 100 [22]. A high score implies a high level of global health status, functioning, and symptoms or problems. In EORTC QLQ-C30, the average scores of the global health scale, functional scales, and symptom scales/items were used to evaluate the “global health QoL”, “functioning QoL” and general cancer symptom-related QoL (“QoL-C30 symptom”), respectively. In EORTC QLQ-HN35, the average score of symptom scales/items was used to assess the QoL of head and neck cancer-related symptoms (“QoL-HN35 symptom”). The pre-IMRT was defined as the period during which NPC patients agreed to participate in this study and before they started the first fraction of IMRT treatment. This period was approximately 1 to 2 weeks. Because significant changes in QoL of NPC patients appear within 2 years after IMRT [23], we evaluated the four scales of QoL levels for each patient at five IMRT-associated time points: pre-IMRT (*n* = 682); during IMRT (around 40 Gy) (*n* = 675), 3 months post-IMRT (*n* = 640); 1-year post-IMRT (*n* = 578); and 2 years post-IMRT (*n* = 505).

### 2.5. Covariates

Demographic and clinical variables were obtained from each NPC patient at the initial clinical treatment. Ethnicity was grouped as Minnan and others (including, Hakka, Mainlander, and aborigines). Educational level was classified as ≤12 and >12 years. Body mass index (BMI) was categorized as underweight (BMI < 18.5), normal weight (BMI = 18.5–23.9), and overweight (BMI ≥ 24.0) according to the criteria of the Taiwan Health Promotion Administration [24]. Comorbidity was classified as 0 and ≥1 comorbid condition, according to Deyo’s Charlson Comorbidity Index (CCI) scoring [25]. The treatment in combination with chemotherapy was recorded as Yes or No.

### 2.6. Statistical Analysis

Proportions were employed to describe the distributions of demographic and clinical factors, and Cox proportional hazards models were used to assess the relationship between these factors and mortality. Overall survival curves were estimated using the Kaplan–Meier method, and the log-rank test was used to evaluate the difference in mortality rates across four clinical stages. We applied mortality density to express the incidence of mortality associated with the clinical stage. Age, gender, ethnicity, educational level, BMI, CCI, chemotherapy, and IMRT treatment period were considered to be potential confounding variables and their effects were adjusted for in all multivariable models. We employed generalized estimating equations with an autoregressive correlation structure to evaluate the influence of advanced stage on QoL scores at different time points. Multivariable Cox proportional hazards models were applied to assess the prognostic effects of different scales of QoL on mortality risks. Adjusted hazard ratios (aHRs) were calculated for every 10-point increase in QoL scores, with the scores ranging from 0 to 100.

Baron and Kenny’s method for identifying mediation was used to assess the possible mediating effect of QoL on the association between advanced tumor stage and patient mortality (Appendix A illustrates the schematic processes) [26]. This approach requires all four conditions have to be met for a mediator. We applied it to our study as follows: (1) the advanced stage was significantly associated with mortality risk; (2) the advanced stage significantly affected specific QoL scales; (3) specific QoL scales significantly affect mortality risk after controlling for advanced stage; and (4) the association between advanced stage and mortality risk was higher than the same association after a specific QoL mediator was adjusted for. The scores of global health QoL, functioning QoL, QoL-C30 symptom and QoL-HN35 symptom at five time points were examined for a mediator. Mediation was measured in effect change, with the excessive effect explained by a specific QoL scale at a time point being calculated as: (aHR1 − aHR2)/(aHR1 − 1),(1)
where aHR1 and aHR2 were the aHRs of advanced stage on patient mortality obtained, respectively, from the base and QoL-adjusted models [27,28,29]. The analysis was performed by using Stata version 16.0 (StataCorp, College Station, TX, USA).

## 3. Results

### 3.1. Demographic and Clinical Characteristics

This cohort included 519 (76.1%) men and 163 (23.9%) women with an average age of 49.4 years (Table 1). The majority of patients were Minnan (80.8%), had ≤12 educational years (72.1%), were overweight (57.9%), had no comorbidity (CCI = 0; 72.5%), received chemotherapy (86.2%), and were treated before 2011 (63.9%). Higher age, being male, less educated, underweight, and treated before 2011 were significantly associated with a higher mortality risk.

### 3.2. Effects of AJCC Stage on Mortality

Table 2 reveals that 9.5%, 24.5%, 35.8%, and 30.2% of NPC patients were diagnosed with AJCC stages I to IV, respectively. Patients with stages III–IV had a significantly higher cumulative mortality rate than did patients with stages I–II (Figure 1; *χ*^2^ = 22.66, *p* < 0.001). The mortality density for NPC patients with stages I to IV was 1.8, 2.5, 3.3, and 8.2 per 100 person-years, respectively. Compared to stage I, patients with stages III and IV had 2.5- and 5.7-fold covariate-adjusted mortality risk, respectively.

### 3.3. Effects of AJCC Stage on QoL Score

The distributions of QoL scores measured for global health QoL, functioning QoL, QoL-C30 symptom and QoL-HN35 symptom at the five time points are shown in Appendix A. Table 3 presents the main and interaction effects of AJCC stage and QoL scores at different time points. Compared with patients with stage I–II at the time point of during IMRT, patients with stage III–IV at pre-IMRT had a significantly lower score of global health QoL and functioning QoL by a large magnitude and a significantly higher score of QoL-C30 symptom and QoL-HN35 symptom by a large magnitude due to the interaction effects of stage and time (*p* ≤ 0.043 for all stage × time interactions). At 3 months and 2 years post-IMRT, the advanced stage had a comparable interaction effect on the scores of global health QoL and QoL-HN35 symptom (*p* ≤ 0.043 for all stage × time interactions) and also at 3 months post-IMRT on the score of QoL-C30 symptom (*p* = 0.002 for stage × time interaction).

Compared to stage I–II patients, stage III–IV patients had a significantly lower covariate-adjusted mean score of global health QoL at pre-IMRT, 3 months and 2 years post-IMRT (51.6 vs. 56.6, 55.1 vs. 60.1, and 63.4 vs. 67.8, respectively; Figure 2A) and a notably lower mean score of functioning QoL at pre-IMRT (82.8 vs. 87.1; Figure 2B). By contrast, advanced-stage patients had a significantly higher mean score of QoL-C30 symptom at pre-IMRT and 3 months post-IMRT (17.5 vs. 13.4 and 19.8 vs. 16.2, respectively; Figure 2C) and a notably higher mean score of QoL-HN35 symptom at pre-IMRT, 3 months and 2 years post-IMRT (17.4 vs. 13.0, 28.6 vs. 24.6, and 21.8 vs. 18.1, respectively; Figure 2D) than did early-stage patients.

### 3.4. Effects of QoL Score on Mortality

Table 4 presents the covariates and AJCC stage-adjusted association between various QoL scores and mortality at the five time points. Patients with a higher level of global health QoL at pre-IMRT and a higher score of global health and functioning QoL at 1 and 2 years post-IMRT exhibited a lower mortality risk (aHR = 0.79–0.92 per 10-point increase; all *p* < 0.05). By contrast, patients having a higher score of QoL-C30 symptom and QoL-HN35 symptom at the abovementioned time points revealed a higher mortality risk (aHR = 1.15–1.34 per 10-point increase; all *p* < 0.05).

### 3.5. Effect Changes Associated with QoL Score Mediation

According to the four requirements of Baron and Kenny’s approach for a mediator, global health QoL, QoL-C30 symptom and QoL-HN35 symptom at pre-IMRT and global health QoL and QoL-HN35 symptom at 2 years post-IMRT exhibited an intermediated effect on the association between advanced stage and high mortality risk (Figure 3). Compared with stage I–II patients, stage III–IV patients had a 2.26-fold covariate-adjusted mortality risk (Table 5, base model). In the base model that was additionally adjusted for global health QoL, QoL-C30 symptom and QoL-HN35 symptom at pre-IMRT, the observed excessive risk was reduced to 2.19-, 2.14-, and 2.14-fold, respectively, with effect changes of 5.8%, 9.8% and 9.3%. By contrast, the scores of global health QoL and QoL-HN35 symptom at 2 years post-IMRT explained 49.4% and 39.4% of the excessive effect of advanced stage on mortality risk (the aHRs reduced to 1.64 and 1.76, respectively, when the base model was additionally adjusted for the two QoL scores).

## 4. Discussion

As far as we know, the study is the first time to demonstrate that QoL acts as a mediator between cancer stage and long-term mortality of cancer patients. For NPC patients, this longitudinal study presents comprehensive findings to demonstrate that AJCC stages III–IV influence patients’ QoL levels at different IMRT-associated time points and that specific QoL scales influence patient mortality. Although the advanced cancer stage in NPC patients significantly predicted high mortality risk, the effects were partially mediated by specific QoL scales at pre-IMRT and 2 years post-IMRT.

NPC patients often exhibit clear debilitating issues of swallowing or hearing and have psycho-social problems related to loss of daily function after treatments, which makes QoL an important outcome evaluation of medical care for this type of cancer [30]. A 14-study systematic assessment on xerostomia-related QoL for IMRT-treated NPC patients revealed that the worst QoL measured in multidimensional scales occurred during or at the end of treatment, but a gradual recovery was observed at 1–2 years after IMRT [23]. Similar findings were recognized in this large-scale NPC patient cohort, in that advanced- and early-stage patients both had worse QoL levels in terms of global health QoL, functioning QoL, QoL-C30 symptom, and QoL-HN35 symptom during IMRT (Figure 2). Our investigation also identified that these four QoL levels gradually improved; however, 3 months post-IMRT, advanced-stage patients had worse global health QoL, QoL-C30 symptom, and QoL-HN35 symptom than early patients and at 2 years post-IMRT they maintained poor global health QoL and QoL-HN35 symptom. The five time trajectory distributions for QoL and the interplay effect of cancer stage and time on the four QoL scales provide empirical information for clinicians to understand the association between IMRT and QoL in advanced- and early-stage NPC patients.

The added value of QoL has been paid ample attention in clinical therapy due to several QoL scales showing independent predictive capability for cancer patient survival [6,7,8,9,10,11]. A meta-analysis of individual patient data from 30 EORTC clinical trials for 11 different cancers suggested that QoL information can help to predict survival for cancer patients [6]. A collective study assessing 17 Canadian randomized controlled trials for eight carcinoma sites supported this argument and indicated that baseline QLQ-C30 QoL scores offer prognostic information in addition to the data from demographic and clinical variables [7]. In a longitudinal study for head and neck carcinoma, specific pretreatment QoL measures, such as dyspnea and appetite loss, were found to link to overall survival [8]. In prospective investigations for NPC patients treated with three-dimensional conformal radiotherapy or IMRT, QoL scales measured at pretreatment and 1 year after treatment have been recognized to have a prognostic value for survival [9,10,11]. Using a comprehensive QoL assessment at five IMRT-related time points, our study further identified that the predictive capability of QoL for NPC patient survival was not restricted to pretreatment and 1 year after IMRT but was extended to 2 years after IMRT, with the strongest prognostic effect of QoL, in fact, being observed at 2 years post-IMRT.

Clinical studies reported that a high proportion of NPC patients were confirmed as AJCC stages III–IV at diagnosis [16,17,18]. Because cancer stage is an unchangeable factor and closely linked to a high mortality [3,4], this highlights the issue of how to prolong survival for advanced-stage NPC patients. QoL in recent studies was recommended as a vital outcome measure in clinical decision-making, especially for advanced-stage cancer patients [6,7,31,32]. Cancer studies using pooled data have indicated that the inclusion of physical functioning, pain and appetite-loss-related QoL in the model with demographic and clinical variables can increase the predictive accuracy by 6% for overall survival of eleven cancers in Europe [6], and the added prognostic value of global health, dyspnea and appetite-loss-related QoL for overall survival of eight cancers was 5% in Canada [7]. Clinical prospective investigations also found that the improvement of QoL within 3 months of treatment was significantly associated with a reduced risk of mortality in colorectal, prostate, and pancreatic cancer patients, even if the related QoL scales were different [33,34,35].

In this study with multiple evaluations for multiple QoL scales over five IMRT-related time points, our findings indicated that 5.8–9.8% of mortality risk for advanced-stage NPC patients was mediated by global health QoL, QoL-C30 symptom, and QoL-HN35 symptom at pre-IMRT. This implies that improvement in these three QoL levels before IMRT treatment is the first step that physicians and families need to help the patients. Our study also revealed that the levels of global health QoL and QoL-HN35 symptom at 2 years post-IMRT mediated a substantial proportion of mortality risk for NPC patients with stages III–IV (39.4–49.4%). This emphasizes that the improvement of global health QoL and QoL-HN35 symptom should continue up to 2 years after IMRT and that this time point could be regarded as a vital QoL checkpoint for monitoring patient survival or tailoring QoL-dependent medical support.

A question in this study is why the mediating effect of QoL on the relationship between NPC stage and mortality significantly manifested 2 years post-IMRT but not at earlier time points. Before IMRT, 3 months post-IMRT, and 2 years post-IMRT, patients with advanced NPC exhibited lower global health QoL and higher QoL-HN35 symptom than did patients with early-stage NPC. However, only the QoL measurements taken pre-IMRT and 2-years post-IMRT were associated with mortality, and a greater effect was observed 2-years post-IMRT than pre-IMRT (Table 4). Moreover, the four QoL measurements taken 1-year post-IMRT were related to mortality, but all QoL scores were nonsignificant between patients in the early and advanced stages (Figure 2). According to the rule established by Baron and Kenny [26], all four mediation conditions have to be met for a variable to be a mediator. Thus, the main mediating effect of QoL manifested 2 years post-IMRT. In a clinical setting, radiation-related late toxicity is a gradual process that is exacerbated with time. In a previous report, the severity of IMRT-related late toxicity was associated with the QoL outcomes of NPC survivors [36]. Therefore, the mortality-mediating effect of QoL 2 years post-IMRT may reflect the late toxicity effect of radiation-related treatment among patients with NPC.

Two strategies for medical treatment and care can improve global health QoL and QoL-HN35 symptom 2 years post-IMRT. The first strategy involves using specific radiotherapy to minimize radiation-related late toxicity. Suggested treatments include the administration of particle or proton therapy, implementation of response-adapted treatment plans following neoadjuvant chemotherapy or during the course of radiotherapy, and reduction in elective nodal volume in certain cases [37]. The second strategy entails enhancing QoL through specific interventions. Recommended interventions include the provision of swallowing function training and oral health care training to reduce the severity of dysphagia [38,39], the substitution of saliva for mouthwash to reduce xerostomia [40], implementation of regular home nursing interventions to improve global health [41], implementation of nutrition support and head-and-neck rehabilitation exercise to ameliorate fatigue [42], implementation of psychological interventions (particularly cognitive behavioral therapy) to mitigate depression or anxiety [43], and implementation of transdisciplinary geriatric and palliative care interventions to enhance the QoL of older adults with cancer [44].

This study has several limitations: first; certain clinical and biological mechanisms may be involved in the association between advanced-stage NPC and high mortality risk, but we only investigated the intermediated effects of QoL from pre-IMRT to within 2 years after IMRT. Second, we did not consider the Epstein-Barr virus because the virus DNA detection data were not fully available in the cohort. Third, the difference in 4 QoL scores for the group comparison at a significant level ranged from 3.6 to 5.0 units, their clinical significances need to be further investigated. Last, 26% of NPC patients were lost to follow-up at 2-years, which may introduce selection bias in the results. Since the distributions of demographic and clinical factors, such as age, sex, ethnicity, body mass index, Charlson comorbidity index, chemotherapy, and IMRT treatment period for the remained and loss patients were comparable, if the bias exists, the degree would be limited. Alternatively, a major strength of this study is that it is the first to elucidate the associations across advanced-stage cancer between multiple health-associated QoL scales over five IMRT-related time points and long-term mortality among NPC patients. Furthermore, the QoL data evaluated at multiple IMRT-associated time points offer clinicians comprehensive information for QoL-dependent medical decisions and care management at the appropriate treatment time.

## 5. Conclusions

We concluded advanced cancer stage correlates with a long-term high mortality in NPC patients treated with IMRT and the association is partially intermediated by QoL at pre-IMRT and 2 years post-IMRT. Therefore, QoL-HN35 symptom and global health QoL-dependent medical support and care should be focused and tailored at 2 years post-IMRT.

## Figures and Tables

**Figure 1 cancers-13-05063-f001:**
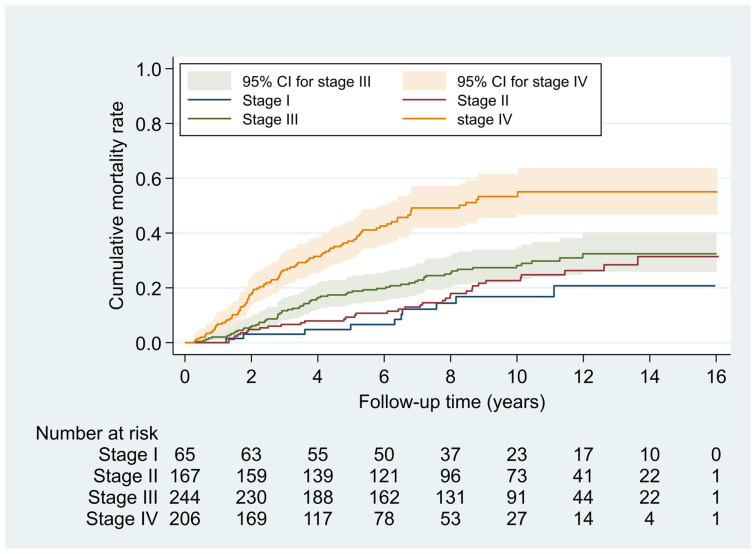
Cumulative mortality rates of nasopharyngeal cancer patients associated with AJCC stages. Note: Cumulative mortality rates were estimated from the Kaplan–Meier estimators. Log-rank test was used to test the equality of cumulative mortality rates between AJCC stages (difference in 4 AJCC stages, *χ*^2^ = 61.65, *p* < 0.001; difference in AJCC stage I–II versus III–IV, *χ*^2^ = 22.66, *p* < 0.001). CI, confidence interval.

**Figure 2 cancers-13-05063-f002:**
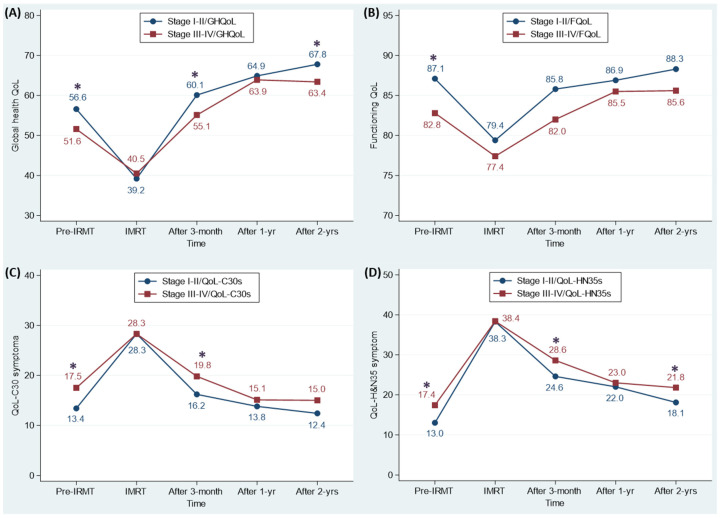
Distributions of adjusted quality of life (QoL) mean scores associated with AJCC stage (I–II/III–IV) at different intensity-modulated radiotherapy (IMRT)-related time points for nasopharyngeal cancer patients. (**A**) Global health QoL; (**B**) Functioning QoL; (**C**) QoL-C30 symptom; (**D**) QoL-HN35 symptom. Note: The investigated time periods included the pre-IMRT, during IMRT (40 Gy), and 3 months, 1 year, and 2 years post IMRT. The QoL scores were adjusted for age, gender, ethnicity, educational level, body mass index, Charlson comorbidity index, chemotherapy, and IMRT treatment period. * denotes significant difference in the adjusted QoL mean scores between AJCC stage I–II and III–IV. GHQoL, Global Health QoL, FQoL, Functioning QoL, QoL-C30 symptom, and QoL-HN35 symptom denoted the average QoL scores obtained from the EORTC QLQ-C30 global QoL scale, 5 functional scales, and 9 symptom scales/items, and EORTC QLQ-HN35 18 symptom scales/items, respectively. IMRT, intensity-modulated radiotherapy; yrs, years; EORTC, European organization for research and treatment of cancer; QoL-C30, the core QoL questionnaire of EORTC; QoL-HN35, the head and neck cancer-specific QoL questionnaire module of EORTC.

**Figure 3 cancers-13-05063-f003:**
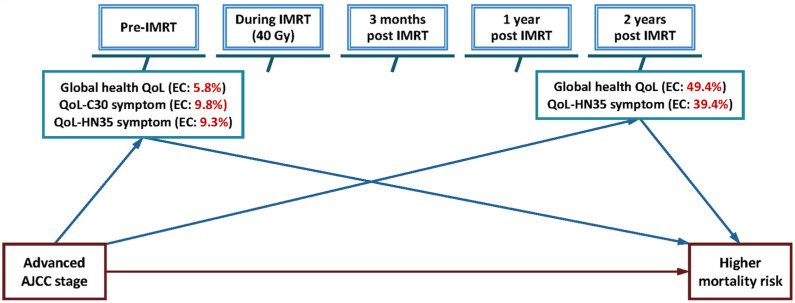
The intermediated paths of quality of life (QoL) scales at different intensity-modulated radiation therapy (IMRT)-related time points on the association between advanced AJCC stage (III–IV) and higher mortality risk in nasopharyngeal cancer patient follow-up cohort (intermediated path: Advanced AJCC stage to QoL score to Mortality risk). Note: Effect change (EC) was the excessive effect explained by a specific QoL scale at a time period. Global health QoL, QoL-C30 symptom, and QoL-HN35 symptom denoted the average QoL scores obtained from the EORTC QLQ-C30 global QoL scale, 9 symptom scales/items, and EORTC QLQ-HN35 18 symptom scales/items, respectively. AJCC, American Joint Committee on Cancer; EORTC, European organization for research and treatment of cancer; QoL-C30, the core QoL questionnaire of EORTC; QoL-HN35, the head and neck cancer-specific QoL questionnaire module of EORTC.

**Table 1 cancers-13-05063-t001:** Distributions and adjusted hazard ratios (aHR) of mortality for demographic and clinical factors in nasopharyngeal cancer patient cohort (*n* = 682).

Factors	No. (%)	aHR ^a^	(95% CI)
**Age, year, mean (SD)**	49.4 (11.5)	-	-
≤40	152 (22.3)	1.0	-
>40	530 (77.7)	1.9	(1.3 to 2.9)
**Gender**			
Female	163 (23.9)	1.0	-
Male	519 (76.1)	2.0	(1.3 to 3.0)
**Ethnicity**			
Minnan	551 (80.8)	1.0	-
Other	131 (19.2)	1.2	(0.9 to 1.7)
**Educational level, year**			
>12	190 (27.9)	1.0	-
≤12	491 (72.1)	1.6	(1.1 to 2.3)
**Body mass index (kg/m^2^) ^b^**			
Normal (18.5–23.9)	268 (39.8)	1.0	-
Underweight (<18.5)	16 (2.4)	2.4	(1.1 to 5.1)
Overweight (≥24.0)	390 (57.9)	1.0	(0.7 to 1.3)
**Charlson comorbidity index**			
0	490 (72.5)	1.0	-
≥1	186 (27.5)	1.2	(0.9 to 1.6)
**Chemotherapy**			
No	94 (13.8)	1.0	
Yes	588 (86.2)	0.7	(0.4 to 1.2)
**IMRT treatment period**			
Before 2011	436 (63.9)	1.0	
2011 onward	246 (36.1)	0.5	(0.3 to 0.7)
**Follow-up time, months**			
Mean (SD)	90.8 (50.8)	-	-
Median (range)	90.5 (3.4–193.0)	-	-

SD, standard deviation; CI, confidence interval; IMRT, intensity-modulated radiation therapy. ^a^ aHRs were adjusted for covariates in the Table and AJCC stage; ^b^ Body mass index groups were categorized according to the criterion of Health Promotion Administration, Ministry of Health and Welfare, Taiwan.

**Table 2 cancers-13-05063-t002:** Mortality densities and adjusted hazard ratios (aHR) of mortality associated with AJCC tumor stage for nasopharyngeal cancer patients after IMRT.

Group	No. of Patients (%)		Years of Follow-Up	No. of Deaths	Mortality Density, per 100 PY	aHR ^a^	(95% CI)
Total	682	(100.0)	5163.1	204	3.95	-	-
AJCC stage							
I	65	(9.5)	583.0	10	1.7	1.0	-
II	167	(24.5)	1468.3	36	2.5	1.8	(0.8 to 3.7)
III	244	(35.8)	1983.9	65	3.3	2.5	(1.2 to 5.3)
IV	206	(30.2)	1127.9	93	8.2	5.7	(2.7 to 12.3)

AJCC, American Joint Committee on Cancer; IMRT, intensity-modulated radiation therapy; PY, person-years; ^a^ aHR was adjusted for age, gender, ethnicity, educational level, body mass index, Charlson comorbidity index, chemotherapy, and IMRT treatment period.

**Table 3 cancers-13-05063-t003:** Main and interaction effect of AJCC stage and quality of life (QoL) scores at different time points.

Factor	Global Health QoL ^a^			Functioning QoL ^a^			QoL-C30 Symptom ^a^			QoL-HN35 Symptom ^a^		
	Adj. β ^b^	(95% CI)	*p*	Adj. β ^b^	(95% CI)	*p*	Adj. β ^b^	(95% CI)	*p*	Adj. β ^b^	(95% CI)	*p*
**Main effect**												
** AJCC stage ^c^**												
I–II	Ref.	-	-	Ref.	-	-	Ref.	-	-	Ref.	-	-
III–IV	1.3	(−2.2 to 4.8)	0.471	−2.1	(−4.5 to 0.4)	0.103	0.08	(−2.2 to 2.4)	0.946	0.08	(−2.3 to 2.4)	0.946
**Time points** ^c^												
Pre-IMRT	17.4	(14.2 to 20.5)	<0.001	7.7	(5.9 to 9.5)	<0.001	−14.9	(−16.6 to −13.1)	<0.001	−25.3	(−27.1 to −23.5)	<0.001
During IMRT (40 Gy)	Ref.	-	-	Ref.	-	-	Ref.	-	-	Ref.	-	-
3 months post IMRT	20.8	(17.7 to 24.0)	<0.001	6.4	(4.6 to 8.2)	<0.001	−12.1	(−13.9 to −10.3)	<0.001	−13.7	(−15.6 to −11.9)	<0.001
1 year post IMRT	25.7	(21.8 to 29.5)	<0.001	7.5	(5.1 to 9.8)	<0.001	−14.4	(−16.7 to −12.1)	<0.001	−16.3	(−18.7 to −14.0)	<0.001
2 years post IMRT	28.6	(24.4 to 32.8)	<0.001	8.8	(6.1 to 11.6)	<0.001	−15.9	(−18.4 to −13.3)	<0.001	−20.2	(−22.9 to −17.5)	<0.001
**Interaction effect (stage × time)**												
III–IV × pre-IMRT	−6.3	(−10.2 to −2.4)	0.001	−2.3	(−4.5 to −0.1)	0.043	4.0	(1.8 to 6.2)	<0.001	4.3	(2.0 to 6.6)	<0.001
III–IV × 3 months post IMRT	−6.3	(−10.3 to −2.3)	0.002	−1.7	(−4.0 to 0.6)	0.143	3.6	(1.3 to 5.8)	0.002	4.0	(1.6 to 6.3)	0.001
III–IV × 1 year post IMRT	−2.3	(−7.1 to 2.5)	0.353	0.7	(−2.3 to 3.7)	0.633	1.2	(−1.7 to 4.1)	0.428	0.9	(−2.1 to 3.9)	0.540
III–IV × 2 years post IMRT	−5.8	(−11.2 to −0.3)	0.038	−0.6	(−4.1 to 2.9)	0.730	2.6	(−0.8 to 5.9)	0.132	3.6	(0.1 to 7.0)	0.043

AJCC, American Joint Committee on Cancer; IMRT, intensity-modulated radiotherapy; Ref., reference group; CI, confidence interval; ^a^ Global health QoL, functioning QoL, QoL-C30 symptom, and QoL-HN35 symptom denoted the average QoL scores obtained from the EORTC QLQ-C30 global QoL scale, 5 functional scales, and 9 symptom scales/items, and EORTC QLQ-HN35 18 symptom scales/items, respectively; ^b^ Adjusted regression coefficients (adj. β) were obtained from generalized estimating equations adjusted for age, gender, ethnicity, educational level, body mass index, Charlson comorbidity index, chemotherapy, and IMRT treatment period; ^c^ Patients with stage I–II and at the time pint of during IMRT (40 Gy) were the reference groups.

**Table 4 cancers-13-05063-t004:** Adjusted hazard ratios (aHR) of mortality associated with continuous quality of life (QoL) scores at different time points of IMRT.

^b^ QoL Scores	Pre-IMRT		During IMRT (40 Gy)		3 Months Post IMRT		1 Year Post IMRT		2 Years Post IMRT	
	aHR ^a^	(95% CI)	aHR ^a^	(95% CI)	aHR ^a^	(95% CI)	aHR ^a^	(95% CI)	aHR ^a^	(95% CI)
Global health QoL	0.92 *	(0.86 to 0.99)	0.94	(0.87 to 1.02)	0.97	(0.89 to 1.05)	0.88 *	(0.80 to 0.97)	0.79 *	(0.69 to 0.91)
Functioning QoL	0.90	(0.82 to 1.00)	0.97	(0.88 to 1.06)	0.93	(0.83 to 1.03)	0.85 *	(0.76 to 0.96)	0.80 *	(0.67 to 0.96)
QoL-C30 symptom	1.15 *	(1.03 to 1.28)	1.00	(0.90 to 1.11)	1.08	(0.97 to 1.21)	1.19 *	(1.05 to 1.35)	1.34 *	(1.11 to 1.63)
QoL-HN35 symptom	1.17 *	(1.05 to 1.31)	1.02	(0.92 to 1.12)	1.02	(0.91 to 1.15)	1.18 *	(1.04 to 1.34)	1.34 *	(1.11 to 1.61)

IMRT: intensity modulated radiotherapy; * *p* < 0.05; ^a^ aHRs were displayed as the risks for every 10-point increase in QoL scores and were adjusted for age, gender, ethnicity, educational level, body mass index, Charlson comorbidity index, chemotherapy, and IMRT treatment period, as well as AJCC stage.; ^b^ Global health QoL, functioning QoL, QoL-C30 symptom, and QoL-HN35 symptom denoted the average QoL scores obtained from the EORTC QLQ-C30 global QoL scale, 5 functional scales, and 9 symptom scales/items, and EORTC QLQ-HN35 18 symptom scales/items, respectively. EORTC, European organization for research and treatment of cancer; QoL-C30, the core QoL questionnaire of EORTC; QoL-HN35, the head and neck cancer-specific QoL questionnaire module of EORTC.

**Table 5 cancers-13-05063-t005:** Adjusted hazard ratios (aHR) of mortality and effect changes in mortality risks associated with AJCC stage and quality of life (QoL) scores at different time points of IMRT.

Models/Variables	Stages III–IV vs. I–II			EC ^a^
	aHR	(95% CI)	*p* Value	
**Base model** ** ^b^ **	**2.26**	(1.56 to 3.27)	<0.001	Ref.
**Base model + QoL scale ^c,d^**				
**Pre-IMRT**				
Global health QoL	2.19	(1.51 to 3.17)	<0.001	5.8%
QoL-C30 symptom	2.14	(1.47 to 3.10)	<0.001	9.8%
QoL-HN35 symptom	2.14	(1.48 to 3.10)	<0.001	9.3%
**2 years post IMRT**				
Global health QoL	1.64	(0.92 to 2.90)	0.091	49.4%
QoL-HN35 symptom	1.76	(0.99 to 3.12)	0.052	39.4%

AJCC, American Joint Committee on Cancer; IMRT, intensity-modulated radiotherapy; Ref., reference group; ^a^ Effect change (EC) was the excessive effect explained by a specific QoL scale at a time period. It was calculated as: [(Base model aHR − QoL-adjusted aHR)/(Base model aHR − 1)] × 100; ^b^ aHR in the base model was adjusted for age, gender, ethnicity, educational level, body mass index, Charlson comorbidity index, chemotherapy, and IMRT treatment period; ^c^ aHRs were obtained from the base model additionally adjusted for a specific QoL scale.; ^d^ Global health QoL, functioning QoL, QoL-C30 symptom, and QoL-HN35 symptom denoted the average QoL scores obtained from the EORTC QLQ-C30 global QoL scale, 5 functional scales, and 9 symptom scales/items, and EORTC QLQ-HN35 18 symptom scales/items, respectively. EORTC, European organization for research and treatment of cancer; QoL-C30, the core QoL questionnaire of EORTC; QoL-HN35, the head and neck cancer-specific QoL questionnaire module of EORTC.

## Data Availability

The data presented in this study are available on request from the corresponding author. The data are not publicly available due to ethical restrictions.

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
