# Peer review of "Quality of Life as a Mediator between Cancer Stage and Long-Term Mortality in Nasopharyngeal Cancer Patients Treated with Intensity-Modulated Radiotherapy"

_cancers, 2021, doi:10.3390/cancers13205063_

Round 1

Reviewer 1 Report

The authors analyzed EORTC QoL surveys of patients with nasopharynx cancer receiving definitive radiotherapy with a 7.5 year median follow-up and found that patients with stage III-IV disease have higher mortality risk vs. stage I-II as expected, and that this risk is partially mediated by global health and head and neck symptom QoL at long term 2-year follow-up. The longitudinal study includes a large cohort of patients (n=682) with comprehensive Qol data, and the manuscript is presented logically. Association between stage and mortality is self-explanatory, and the association between QoL and mortality in cancer patients has been studied extensively in recent years. Novelty of the study comes from exploring QoL as a mediator of the influence of stage on mortality.

Comments:

-IMRT technique was shown to be significantly associated with mortality, namely a mortality risk reduction by half for the arc vs. fixed. We would not expect radiation technique to influence mortality in this setting, and this finding should be explained in detail. Did the difference in the treatment era influence the patient population, other treatment parameters, etc.?

-It is known that stage and QoL influence mortality. Why would the mediating effect of QoL on the relationship between stage and mortality manifest at a specific time point (2 years) and not others? It would be helpful to add the authors’ thoughts on how this might be plausible (i.e. why it may not be by chance). 

-Please add a discussion on what can be done clinically at long term follow-up that could potentially improve the patient’s global and head and neck symptom associated QoL as the authors suggest should be done.

Minor comments:

-Information regarding chemotherapy (page 3 lines 101-106) is repeated (lines 106-110) and should be deleted.

-There was a 26% loss of follow-up at 2 years (682 patients to 505 patients), which can introduce bias and should be discussed as a limitation.

-Please clarify what is meant by “enhanced” score (page 6)

Author Response

Response to the comments from reviewer #1:

-IMRT technique was shown to be significantly associated with mortality, namely a mortality risk reduction by half for the arc vs. fixed. We would not expect radiation technique to influence mortality in this setting, and this finding should be explained in detail. Did the difference in the treatment era influence the patient population, other treatment parameters, etc.?

Response: Thanks for the valuable comments. We agree with the argument that the reviewer raised. As shown in the “Treatment” section, the fixed-beam IMRT with two-phase sequential planning was applied before 2011; and the rotational-arc IMRT with simultaneously integrated boost planning was applied after 2011 in the institute. The survival difference between the two groups might be attributed to some unadjusted variables, such as the examination tool of PET-CT, plasma EBV-DNA, and the type of drug therapy that was available in the institute in the era before or after 2011. In the result section, we just mentioned that the fixed-beam IMRT was significantly “associated with” a higher mortality risk, not emphasizing risk level, here, 2.0-fold risk (reciprocal of 0.5), please see lines 209-210. In the limitation section, we added a statement to explain this issue, please see the highlights, page 13 lines 400-403.

-It is known that stage and QoL influence mortality. Why would the mediating effect of QoL on the relationship between stage and mortality manifest at a specific time point (2 years) and not others? It would be helpful to add the authors’ thoughts on how this might be plausible (i.e. why it may not be by chance). 

Response: Thanks for the kind comment. As we known, radiation-related late toxicity is a gradual process and has a greater effect in the late stage. In previous studies, the QoL data were usually collected at one-specific time point, such as before IMRT treatment, or 3 or 6-months and 1-year after IMRT treatment that prevents the complete evaluation for the effect of QoL on head and neck cancer or NPC patient mortality (Ediebah DE, et al. Cancer 2018; Yang CJ, et al. Qual Life Res 2016; Tsai WL, et al. Qual Life Res 2013; Oskam IM, et al. Radiother Oncol 2010). In our previous report, the severity of radiation-related late toxicities was observed to be associated with the QoL outcome for NPC survivors (Tsai WL, et al. BMC Cancer 2014). The QoL after 2-years of IMRT may reflect the late toxicity effect of radiation-related treatment among NPC patients. Thereby, we speculated that the QoL assessed at more years after IMRT might have a mediating effect on NPC mortality. We have added the related statement in the Discussion, please see the highlights on page 12 Lines 384-390.

-Please add a discussion on what can be done clinically at long term follow-up that could potentially improve the patient’s global and head and neck symptom associated QoL as the authors suggest should be done.

Response: Thanks for the suggestion. To improve the global and head and neck symptom associated QoL during the long-term follow-up period, an interventional plan is needed, which might include a regular home nursing intervention (Shi RC, et al. Asian Pac J Cancer Prev 2015), a nutrition counseling combined with head and neck rehabilitation exercise (Su D, et al. Ann Palliat Med 2020), or a psychological intervention, especially cognitive behavioral therapy (Liu F, et al. Integrative Cancer Therapies 2021). We have added these information in the Discussion, please see the highlights on page 12 Lines 391-395.

Minor comments:

-Information regarding chemotherapy (page 3 lines 101-106) is repeated (lines 106-110) and should be deleted.

Response: The redundant statements have been deleted. Thanks for the suggestion.

-There was a 26% loss of follow-up at 2 years (682 patients to 505 patients), which can introduce bias and should be discussed as a limitation.

Response: Thanks for the comment. For this issue, we have checked the distributions of demographic and clinical factors for the remained and loss patients. Our data showed that the distribution of age, sex, ethnicity, body mass index, Charlson comorbidity index, chemotherapy, and IMRT technique were all comparable (P for chi-squared test: 0.155, 0.335, 0.652, 0.157, 0.942, 0.264, 0.220, and 0.942, respectively). However, this is an important issue for the validity of the study results. We have added it in the study limitation, please see page 13 Lines 405-410.

-Please clarify what is meant by “enhanced” score (page 6)

Response: The term “enhanced” was used to express the interaction effect. However, we seem to provide an unclear statement for the results. We have modified the related statements, please see the highlights on page 7 Lines 236-243. Many thanks for the comment.

Reviewer 2 Report

Thank you for the opportunity to review this article. Please find my specific comments below.

  1. The number of individuals who completed the QoL questionnaires are decreasing overtime. (line 146-147). Did those patients die or could not complete the questionnaire due to health conditions? The authors emphasized the importance of QoL at 2 years. But this would require patients to have survived up to that point. Those who completed the 2-year QoL questionnaire may be healthier than those who did not. It is important to account for patients missing questionnaires. Otherwise, the estimates may be biased.
  2. What is the pre-IMRT period?
  3. Treatment dose gradually increased to 40 Gy and 3 months post IMRT were listed as one period. Are all patients achieved this in 3 months post IMRT? (line 146)
  4. Do all four conditions have to be met for a variable to be a mediator? Lines 174-178.
  5. Are there any theory/methodology paper showing that the % change in adjusted HRs can be used to measure the mediation effect? The cited papers 27-29 included two papers written by the authors.
  6. Lines 106-109 seem to repeat the info in lines 101-105. Please check.
  7. Lines 218-2224. The results reported here are inconsistent with those in the Table 3. Please check. Also “enhanced” lower or higher score are hard to understand. Please consider alternative ways to describe the results.
  8. Figure 2. Although there are some differences in QoL measures, the differences are mostly 2-3 units. Are these considered minimal clinically significant?
  9. 5 Effect changes associated with OoL section. It is not clear how the authors arrived at the conclusion that 2 years post exhibited an intermediated effect but not 1 year post. From Table 4, 1-year post had similar effect as 2-year post.
  10. Please spell out EORTC on page 3 if it has not been referenced before.
  11. Please add discussion about what health care provider could do with this information. While knowing this is important, what can be done to reduce mortality?

Author Response

Response to the comments from reviewer #2:

  1. The number of individuals who completed the QoL questionnaires are decreasing overtime. (line 146-147). Did those patients die or could not complete the questionnaire due to health conditions? The authors emphasized the importance of QoL at 2 years. But this would require patients to have survived up to that point. Those who completed the 2-year QoL questionnaire may be healthier than those who did not. It is important to account for patients missing questionnaires. Otherwise, the estimates may be biased.

Response: Many thanks for the comments. Because the mortality rate from NPC is low (5 year mortality rate around 25%), the patients who cannot survive to the second years after IMRT treatment was relatively low. However, there was a 26% of NPC patients loss to follow-up at 2 years, which is an issue of selection bias. For this issue, we have checked the distributions of demographic and clinical factors for the remained and loss patients. Our data showed that the distribution of age, sex, ethnicity, body mass index, Charlson comorbidity index, chemotherapy, and IMRT technique were all comparable (P for chi-squared test: 0.155, 0.335, 0.652, 0.157, 0.942, 0.264, 0.220, and 0.942, respectively). However, this is an important issue for the validity of the study results. We have added it in the study limitation, please see page 13 Lines 405-410.

  1. What is the pre-IMRT period?

Response: The pre-IMRT period was defined as the period during NPC patients agreed to participant in this study and before they started the first fraction of IMRT treatment. The pre-IMRT period was approximately 1 to 2 weeks. We have added this information to the revised manuscript. Please see lines 156-158. Thanks for the comment.

  1. Treatment dose gradually increased to 40 Gy and 3 months post IMRT were listed as one period. Are all patients achieved this in 3 months post IMRT? (line 146)

Response: This is a redundant statement. We are sorry about that! The four scales of QoL levels were evaluated at 5 IMRT-associated time points (pre-IMRT, during IMRT (around 40Gy), 3 months post-IMRT, 1 year post-IMRT, 2 years post-IMRT). The redundant statement has been deleted in the revised manuscript. Please see the highlights on lines 160-162.

  1. Do all four conditions have to be met for a variable to be a mediator? Lines 174-178.

Response: Yes, all four conditions have to be met for a variable to be a mediator. We have offered this criterion in the “Statistical analysis”. Please see lines 195-196.

  1. Are there any theory/methodology paper showing that the % change in adjusted HRs can be used to measure the mediation effect? The cited papers 27-29 included two papers written by the authors.

Response: Thanks for the comment. We added a theory/methodology related paper for this issue (Baron and Kenny, J Pers Soc Psychol, 1986). The cited papers #27-29 are the practical applications using the Baron and Kenny’s methodology for mediation effects. Please see line 201 and reference #26.

  1. Lines 106-109 seem to repeat the info in lines 101-105. Please check.

Response: The redundant statements have been deleted. Thanks for the suggestion.

  1. Lines 218-224. The results reported here are inconsistent with those in the Table 3. Please check. Also “enhanced” lower or higher score are hard to understand. Please consider alternative ways to describe the results.

Response: Many thanks for the comment. The term “enhanced” was used to express the interaction effect. However, we seem to provide an unclear statement for the results. We have modified the related statements; please see the highlights on page 7 Lines 236-243.

  1. Figure 2. Although there are some differences in QoL measures, the differences are mostly 2-3 units. Are these considered minimal clinically significant?

Response: Many thanks for the valuable comment. Please see Figure 2, the difference in 4 QoL scores for the group comparison at significant level (denoted as ‘*’) ranged from 3.6 to 5.0 units, because of the interaction between stage and time. Here, the QoL scores are an average score of all item-specific scores in 4 domains, thus their clinical significances need to be further investigation. We have added this point in the limitation section for this manuscript. Please see the highlights on page 13 Lines 403-405.

  1. 5 Effect changes associated with QoL section. It is not clear how the authors arrived at the conclusion that 2 years post exhibited an intermediated effect but not 1 year post. From Table 4, 1-year post had similar effect as 2-year post.

Response: Thanks for the comment. Please see lines 187-196 in the ‘Statistical analysis’ section. According to the Baron and Kenny’s method, there are 4 statistical procedures for identifying mediation. All of the 4 conditions have to be met for a variable to be a mediator. In Table 5, only the QoL scores that met the 4 conditions have been discussed their mediated effects. Please see Figure 2, the differences in the 4 QoL scores at 1 year post between III/IV and I/II stages were not significant. For clarifying this issue, we have added this criterion in the revised manuscript. Please see lines 195-196.

  1. Please spell out EORTC on page 3 if it has not been referenced before.

Response: Thanks for the suggestion. The full name for EORTC has been spell out in the new version of manuscript. Please see lines 46-47.

  1. Please add discussion about what health care provider could do with this information. While knowing this is important, what can be done to reduce mortality?

Response: Thanks for the suggestion. For this issue, we have added the following statements in Discussion section, please see highlights on page 12 Lines 391-395.

“To improve the QoL for global health and head and neck symptom during the long-term follow-up period, an interventional plan is needed, which might include a regular home nursing intervention, a nutrition counseling combined with head and neck rehabilitation exercise, or a psychological intervention, especially cognitive behavioral therapy.”

Round 2

Reviewer 1 Report

-The updates to the manuscript are overall cursory in nature, and does not fully address the comments. The added content has suboptimal readability and grammatically inconsistent, and as a result, fails to enhance the manuscript. Edits, including the simple summary, should be re-written so that the language is concise and the style is in line with the original manuscript.

-RT technique affecting survival to the magnitude demonstrated in the data affects the credibility of the dataset. The unaccounted variables such as concurrent chemotherapy, prevalence of PET/CT, etc. that the authors mention need to be investigated and actual details provided (e.g. differences in chemotherapy agents used, use of induction chemo, stage migration, etc.), to make this a plausible finding.

-The reason for why a 2 year follow-up, and not other time points, acts as a mediator of the relationship between stage and mortality is not adequately explained. Yes, the 2 year follow-up represents late QOL, but why late QOL mediates survival and not earlier QOLs is not discussed. Other studies that collected data at a more limited number of or earlier time points is not relevant for this discussion because these studies are not looking at the mediating effects of QOL in the relationship between stage and mortality.

-Strategies to enhance patient’s global and H&N specific QOL need to be expanded to include published data on way certain interventions can specifically enhance QOL, e.g. speech language pathology and swallowing function, palliative care involvement and chronic pain, dental care and oral health, oral rinses/lozenges/sprays and xerostomia, geriatric care on global functioning of older patients, etc. in addition to nutrition and psychologic support mentioned. Comments should also be made about how it’s important to design the radiation up front to minimize late effects on patient’s QOL. The added value of the manuscript is that it can inform clinicians of the importance of QOL assessment years after completing RT and how responding to it can impact outcomes. Without a detailed discussion of how this can be done, the message has less impact.

-Enhanced score has been changed to intensively lower/higher, but the message is still difficult to comprehend. I would suggest language such as “had a significantly lower score by a large magnitude” if that is the intent of the statements. Also, the language needs to be updated in the abstract.

Reviewer 2 Report

Thank you for the authors responses. Below please see a few additional comments.

  • Response: Yes, all four conditions have to be met for a variable to be a mediator. We have offered this criterion in the “Statistical analysis”. Please see lines 195-196.

Please make it clear in the section that all 4 conditions have to be met.

  •  Response: Thanks for the suggestion. For this issue, we have added the following statements in Discussion section, please see highlights on page 12 Lines 391-395.

“To improve the QoL for global health and head and neck symptom during the long-term follow-up period, an interventional plan is needed, which might include a regular home nursing intervention, a nutrition counseling combined with head and neck rehabilitation exercise, or a psychological intervention, especially cognitive behavioral therapy.”

Thanks for the addition in the discussion. However, it only listed potential interventions. There were no references provided that showed these interventions are effective in improving QoL in head and neck cancer patients. Please revise and expand this discussion.
